# Impact of Dietitian-Guided Individualized Nutrition (DGIN) on ICU Outcomes in Critically Ill Patients: A Retrospective Cohort Study in Taiwan

**DOI:** 10.3390/nu17182995

**Published:** 2025-09-18

**Authors:** Shih-Ching Lo, Hsing-Chun Lin, Yu-Hsun Wang, Ying-Ru Chen, Shun-Fa Yang

**Affiliations:** 1Institute of Medicine, Chung Shan Medical University, Taichung City 402, Taiwan; mygodway@gmail.com; 2Department of Nutrition, Chung Shan Medical University Hospital, Taichung City 402, Taiwan; cshc143@csh.org.tw (H.-C.L.); cshc195@csh.org.tw (Y.-R.C.); 3Department of Nutrition, Chung Shan Medical University, Taichung City 402, Taiwan; 4Department of Medical Research, Chung Shan Medical University Hospital, Taichung City 402, Taiwan; cshe731@csh.org.tw

**Keywords:** critical care, nutrition support, dietitians, length of stay, dietitian-guided individualized nutrition, clinical outcomes

## Abstract

**Background:** On 1 October 2019, the Taiwan National Health Insurance (NHI) Administration introduced reimbursement for nutritional care services provided to intensive care unit (ICU) patients, under the category of nutritional care fees. These services included the implementation of structured, dietitian-guided individualized nutrition (DGIN) protocols designed to address the clinical needs of critically ill patients. **Objectives:** This study aimed to evaluate the effectiveness of DGIN in critically ill patients following the implementation of NHI coverage. **Methods**: This retrospective cohort study was conducted in the ICU of a tertiary medical center, including patients admitted between 1 September 2018 and 31 October 2020, encompassing periods both before and after the initiation of NHI coverage. Under NHI coverage, dietitian-guided individualized nutritional support was provided within the first two weeks of admission. The intervention group received DGIN within the first two weeks of ICU admission. A total of 5292 patients were screened; 2381 were included in the final analysis (1116 in the standard care (SC) group and 1265 in the DGIN group), categorized based on the timing of NHI coverage and the corresponding frequency of dietitian visits. The DGIN protocol comprised a baseline assessment within 24–48 h and three structured reviews during the first ICU week, while the comparator received SC. Demographic characteristics, daily nutritional data, and clinical outcomes were analyzed. **Results:** Significant baseline differences in nutritional intake and disease severity were observed. Following the introduction of the DGIN protocol, the intervention group received more structured and closely monitored nutrition management, which resulted in less aggressive caloric intake. This approach was associated with a significantly shorter ICU length of stay (SC: 8.1 ± 6.7 days vs. DGIN: 7.1 ± 7.4 days, *p* < 0.001). **Conclusions**: An ICU nutritional care plan involving frequent assessments and timely interventions by clinical dietitians is associated with a reduced ICU length of stay in critically ill patients. These findings support the effectiveness of integrating dietitian-led nutritional care into national health insurance coverage for ICU patients.

## 1. Introduction

### 1.1. Critical Care Nutrition

Nutrition is integral to the management of critically ill patients [1]. The European Society for Clinical Nutrition and Metabolism (ESPEN) recently updated its guidelines to provide evidence-based recommendations for nutritional care in the intensive care setting [2]. The Early Versus Delayed Enteral Feeding to Treat People With Acute Lung Injury or Acute Respiratory Distress Syndrome (EDEN) randomized trial compared trophic enteral feeding (~20 kcal/h) with full enteral feeding (25–30 kcal/kg/day) for up to six days. Although ventilator-free days, 60-day mortality, and infectious complications did not differ, trophic feeding was associated with fewer episodes of gastrointestinal intolerance [3]. Similarly, the Augmented versus Routine Approach to Giving Energy (TARGET) trial showed that delivering approximately 100% of estimated caloric requirements, compared with ~70% [4], during critical illness did not improve mortality, health-related quality of life, or functional outcomes [5]; the PermiT randomized trial likewise showed no difference in 90-day mortality between permissive underfeeding and standard feeding [4]. However, a substantial proportion of intensive care unit (ICU) patients continue to receive suboptimal nutritional support [6]. Notably, participants in these landmark trials were predominantly well nourished and overweight, with a mean body mass index (BMI) near 30 kg/m^2^. Moreover, moderate obesity has been associated with improved outcomes during critical illness—a phenomenon often termed the “obesity paradox” [7,8]. Critically ill patients mount a profound stress response with accelerated catabolism, loss of lean body mass, and frequent feeding intolerance, placing them at high risk of malnutrition, weakness, and delayed recovery [9]. Contemporary guidance generally supports early enteral nutrition when feasible and individualized energy/protein prescriptions while avoiding overfeeding; however, the optimal timing, dose, and escalation strategy remain debated [10]. In practice, delivery is challenged by hemodynamic instability, gastrointestinal intolerance, frequent interruptions for procedures, heterogeneous use of predictive equations versus indirect calorimetry, and inconsistent documentation—factors that can create both under- and over-feeding windows [11]. These gaps are exacerbated by interruptions to enteral nutrition delivery that accrue energy deficits [12,13], and by multicenter observations that energy and protein delivery often fall short of targets in routine ICU care [14,15]. Structured, dietitian-guided individualized nutrition (DGIN) protocols standardize assessment and adjustment via prespecified reassessments and algorithmic changes in route, rate, formula, and adjuncts based on tolerance, organ function, glycemic control, and inflammatory status. Real-world evidence on such protocols, particularly in Asian ICUs, remains limited. We therefore compared a DGIN program with standard care (SC) in a single-center historical-control cohort, focusing on index ICU length of stay, in-hospital mortality, and early hospital readmission (14- and 30-day).

### 1.2. The Taiwan National Health Insurance (NHI) Program

Since the implementation of Taiwan’s National Health Insurance (NHI) program in March 1995, there has been a significant reduction in patient mortality and an improvement in the overall quality of medical care [16]. With improvements in healthcare delivery, the efficient allocation of resources has gradually become a critical issue. Due to expenditures consistently exceeding revenues, the NHI system has undergone several structural reforms. Since 2013, the adoption of a global budget allocation mechanism has contributed to a gradual balance between healthcare demand and supply [17]. Consequently, researchers have increasingly focused on the efficiency of healthcare service allocation to ensure the sustainability of the NHI program. Numerous studies have examined the evolving trade-offs between quality and efficiency [18], as well as the importance of reinforcing financial management and quality control systems [19]. In response to these challenges, the Taiwan National Health Insurance Administration revised its reimbursement policies accordingly [20]. On 1 October 2019, the NHI Administration introduced a new reimbursement category for nutritional care services provided to ICU patients, reflecting the growing recognition of clinical nutritional needs [21]. Previous studies have reported a 35.8% reduction in costs per patient following the implementation of early nutrition therapy [22]. Our previous research further identified adequate nutritional intake as a key determinant in facilitating successful ventilator weaning [23]. To enhance the effectiveness of NHI policies and emphasize the critical role of clinical dietitians in ICU nutrition, this study aimed to evaluate the clinical impact of structured dietitian-guided individualized nutrition (DGIN) in critically ill patients under Taiwan’s NHI program. Recent ICU research has revealed a paradigm shift from traditional standardized enteral feeding to more individualized strategies [24]. In particular, the early administration of high-protein nutrition and excessive caloric intake may have detrimental effects. Personalized nutritional support, tailored to each patient’s clinical condition, is now considered superior to one-size-fits-all protocols.

## 2. Materials and Methods

### 2.1. Study Population and Setting

Study design and scope. This single-center, observational retrospective cohort was designed to estimate associations between DGIN and clinical outcomes compared with standard care (SC); it was not intended to establish causality. The study was conducted in the intensive care units (ICUs) of a medical center from 1 September 2018 to 31 October 2020, spanning the periods before and after implementation of Taiwan’s National Health Insurance (NHI) coverage for new ICU nutritional care services. Participating units included the surgical ICU (SICU), medical ICU (MICU), coronary care unit (CCU), and burn ICU (BICU). The neonatal ICU (NICU), pediatric ICU (PICU), and the respiratory care center (RCC) were excluded.

A total of 5292 ICU admissions were screened and categorized into two groups according to both era and treatment actually received: patients managed with the structured DGIN protocol (post-implementation period with protocol adherence) and those receiving SC (pre-implementation period without DGIN). To minimize potential confounding, we excluded patients who had a pre-existing dietitian-managed plan with stable nutritional intake before ICU admission, those aged < 18 or >90 years, and those with extreme body mass index (BMI) values (<15 or >40 kg/m^2^). Baseline demographics, nutritional intake, laboratory measurements, and clinical outcomes were collected from the electronic medical record.

DGIN protocol. An overview of the cohort is shown in Figure 1; the DGIN assessment schedule is summarized in Figure 2. In the DGIN group, a structured intervention was implemented, beginning with a dietitian assessment within 48 h of ICU admission, followed by two reassessments over the next five days and three additional evaluations in the subsequent week, with further reviews as clinically indicated. This schedule supported frequent, individualized adjustments to energy and protein prescriptions and the route of feeding.

Outcome definitions. The primary outcome was ICU length of stay (LOS), defined as the number of days from ICU admission to ICU discharge or in-ICU death during the index ICU admission; days accrued after any subsequent ICU readmission were not added to the index LOS.

Readmission outcomes. Hospital readmission was defined as any unplanned readmission to an inpatient unit (ward or ICU) within 14 and 30 days after hospital discharge from the index hospitalization; analyses were restricted to patients discharged alive.

Ethics. The study complied with the Declaration of Helsinki and was approved by the Chung Shan Medical University Hospital Institutional Review Board (IRB No. CS1-22184).

### 2.2. Study Groups and Nutrition Protocols

Patients were stratified into two groups based on the prevailing nutritional care model during their ICU admission.

The DGIN group received a structured, protocol-driven program. In this model, a registered dietitian performed a comprehensive baseline assessment within 48 h of ICU admission, followed by two scheduled reassessments during the first week (e.g., ICU days 3 and 5) and three additional follow-up evaluations from the second week onward (timing flexible according to clinical indication), after which reassessments continued as clinically indicated for the remainder of the ICU stay. At each visit, the dietitian adjusted energy and protein targets, feeding routes, administration rates, and the use of adjuncts based on the patient’s evolving clinical status, including gastrointestinal tolerance, organ function, glycemic control, and inflammatory state.

The SC group represented the standard institutional care, which consisted of routine dietitian-led nutritional follow-ups every three working days. An initial visit was performed within one working day of ICU admission, followed by routine documentation at three-working-day intervals. This process was not triggered by physician consults but was part of the baseline institutional protocol.

Sensitivity analysis for confounding. To address baseline imbalance, we performed a priori 1:1 nearest-neighbor propensity-score matching (caliper 0.2) using the following baseline covariates: sex, age, admission type (medical vs. surgical), BMI, APACHE II score, serum albumin, and initial daily energy and protein intake. Outcomes were re-estimated in the matched cohort; detailed results are provided in Appendix A.

### 2.3. Statistical Analysis

Continuous variables are presented as mean ± standard deviation (SD), and categorical variables as counts and percentages. Between-group comparisons of continuous variables were performed using the independent *t*-test, while categorical variables were analyzed using the Pearson χ^2^ test. The Mann–Whitney U test was applied to compare non-normally distributed continuous variables, such as ICU length of stay. Two-sided *p* values < 0.05 were considered statistically significant. Logistic regression was used to assess the association between nutritional intervention and binary clinical outcomes (mortality and hospital readmission), with adjustment for clinically relevant covariates, including body mass index (BMI), APACHE II score, energy delivery, protein intake, albumin, and potassium. Results were expressed as odds ratios (ORs) with 95% confidence intervals (CIs). Kaplan–Meier survival analysis was used to compare in-hospital survival between the DGIN and SC groups, and differences were assessed using the log-rank test. Analyses were conducted on complete cases without imputation. All analyses were performed using SPSS Statistics version 20.0 (IBM Corp., Armonk, NY, USA).

### 2.4. Propensity Score Matching Analysis

To address potential confounding and selection bias inherent in this observational study, we performed a sensitivity analysis using propensity score matching (PSM). Propensity scores, representing the probability of a patient being in the DGIN group, were estimated using a logistic regression model. The model included the following eight baseline covariates: sex, age, admission type (medical vs. surgical), BMI, APACHE II score, serum albumin, and initial daily energy and protein intake.

We then created a matched cohort using a 1:1 nearest neighbor matching algorithm without replacement, with a caliper of 0.2 standard deviations of the logit of the propensity score. The balance of covariates after matching was assessed using *p* values, as shown in Appendix A. All PSM analyses were performed using SAS software (Version 9.4, SAS Institute Inc., Cary, NC, USA).

## 3. Results

### 3.1. Study Population and Baseline Characteristics

A total of 2468 and 2824 patients were initially assigned to the DGIN and SC groups, respectively. Exclusions were applied for the following reasons: ICU stay of less than 48 h (DGIN: 577; SC: 1143), admission to non-general adult ICUs (DGIN: 471; SC: 487), transfer before treatment completion (DGIN: 35; SC: 22), refused treatment or received palliative care (DGIN: 12; SC: 8), missing nutritional intake records (DGIN: 101; SC: 42), and patients aged under 18 years (DGIN: 7; SC: 6). After applying these criteria, 1265 patients in the DGIN group and 1116 in the SC group were included in the final analysis (Figure 1). However, significant differences in nutritional management emerged following the implementation of the treatment protocol in the ICU. The baseline characteristics of patients at enrollment are summarized in Table 1. Notable baseline differences between the groups were observed in BMI, body weight, APACHE II score, and albumin levels—clinically relevant variables that could potentially influence outcomes. Although these imbalances were inherent to real-world conditions, statistical adjustments were applied in the analysis of clinical outcomes to account for these baseline differences.

### 3.2. Nutritional Intake and Biochemical Indicators

Significant differences in nutritional intake were observed between the two groups. Upon ICU admission, both energy and protein intake were slightly but significantly lower in the DGIN group compared to the SC group (energy: 13.4 ± 10.5 vs. 14.7 ± 11.1 kcal/kg/day, *p* = 0.005; protein: 0.5 ± 0.4 vs. 0.6 ± 0.5 g/kg/day, *p* < 0.001). During the final stable phase of ICU care (defined as the period immediately prior to ICU discharge), the average energy and protein intake in the DGIN group remained significantly lower than in the SC group (energy: 18.0 ± 10.4 vs. 19.6 ± 10.1 kcal/kg/day, *p* < 0.001; protein: 0.7 ± 0.4 vs. 0.8 ± 0.5 g/kg/day, *p* < 0.001). No significant difference was found in body weight change during the ICU stay (*p* = 0.828); however, post-intervention BMI was slightly but significantly higher in the DGIN group (24.4 ± 9.4 vs. 23.5 ± 7.0 kg/m^2^, *p* = 0.002). While serum albumin levels were significantly higher in the DGIN group at baseline, this difference did not remain statistically significant post-intervention in the more robust propensity score-matched analysis. Other biochemical markers, including prealbumin, hemoglobin, renal function indices (BUN and creatinine), and inflammatory markers (C-reactive protein), did not differ significantly between groups.

### 3.3. Clinical Outcomes

The DGIN group had a significantly shorter ICU length of stay than the SC group (7.1 ± 7.4 vs. 8.1 ± 6.7 days; *p* < 0.001; Figure 3A). There was no significant difference in in-hospital mortality between groups (19.1% vs. 21.6%; *p* = 0.123), consistent with the Kaplan–Meier analysis, which also showed no difference in survival over time (log-rank *p* = 0.124; Figure 3B). However, both 14-day and 30-day hospital readmission rates were higher in the DGIN group (14-day: 24.2% vs. 12.0%; *p* < 0.001; 30-day: 32.1% vs. 13.7%; *p* < 0.001), including in models adjusted for prespecified clinical and nutritional covariates (Table 2). In the propensity-score–matched cohort, index ICU LOS was 7.4 ± 7.6 vs. 8.1 ± 6.8 days (DGIN vs. SC; *p* < 0.001); in-hospital mortality did not differ (20.6% vs. 20.5%; *p* = 0.123); and 14-/30-day hospital readmission was higher with DGIN (adjusted OR 2.42, 95% CI 1.61–3.63; adjusted OR 3.05, 95% CI 2.01–4.62).

## 4. Discussion

### 4.1. Summary of Key Findings

This study evaluated the impact of a DGIN protocol on critically ill patients admitted to the ICU. Compared to patients receiving SC, those in the DGIN group demonstrated a significantly shorter ICU length of stay and modestly improved post-ICU nutritional indicators. However, in-hospital mortality did not differ significantly between the groups, and the DGIN group unexpectedly exhibited higher 14-day and 30-day hospital readmission rates. These findings suggest that while structured nutritional support may facilitate improved metabolic stabilization and earlier ICU discharge, its effects on long-term outcomes remain complex. Similar trends have been reported in critically ill, oncology, and older patient populations, where disease severity, functional decline, and other post-discharge factors may mitigate the impact of nutritional interventions alone [25,26,27]. These observations highlight the need for integrated, multidisciplinary strategies beyond nutrition to improve outcomes in high-risk populations.

### 4.2. ICU Length of Stay and Nutritional Stabilization

Our finding that the DGIN protocol significantly reduced ICU length of stay aligns with prior evidence suggesting that early or enhanced nutritional support may facilitate faster recovery in critically ill patients. In this study, the shortened ICU stay observed in the DGIN group may reflect the impact of more individualized and adaptive nutritional planning, wherein structured and frequent dietitian involvement enabled timely adjustments to patients’ evolving metabolic needs and clinical status [28].

Although the DGIN group received slightly lower average energy and protein intakes compared to the SC group, nutritional care was delivered with greater frequency and responsiveness, potentially preventing overfeeding during the acute catabolic phase and more accurately matching each patient’s physiological condition. This supports the idea that the precision and timing of nutritional support, rather than absolute intake values, may be more critical in influencing recovery. This interpretation is consistent with the 2023 ESPEN guidelines, which emphasize that early identification of nutritional risk and individualized, regularly reassessed interventions in hospitalized patients with multimorbidity are associated with improved clinical outcomes and reduced length of hospital stay [29]. Nevertheless, caution is warranted in attributing causality solely to the nutritional intervention. Other factors such as ICU discharge criteria, variations in care team practice, or unmeasured confounders may have influenced length of stay. Furthermore, previous studies have shown that early hypocaloric or permissive underfeeding may not impact mortality or complication rates, despite potentially shortening ICU stays [30]. These observations are concordant with recent randomized trials (e.g., EFFORT Protein, PRECISe, TARGET Protein, and NUTRIREA-3), in which higher early protein/calorie delivery did not improve major clinical outcomes in general ICU populations [31,32,33,34]. Together, these findings underscore the potential value of structured, protocol-driven nutritional care—such as DGIN—in supporting metabolic stabilization and optimizing ICU resource utilization. Still, further prospective studies are needed to clarify its independent effect and to integrate it effectively within a comprehensive, multimodal critical care strategy.

### 4.3. Unexpectedly Higher Readmission Rates

While structured nutritional support protocols such as DGIN may reduce ICU length of stay, the higher 14- and 30-day readmission rates observed in the DGIN group were unexpected. Emerging evidence suggests that higher early caloric delivery during the initial days of ICU care has been associated with a greater risk of subsequent readmission. For instance, in patients with acute heart failure, a caloric intake ≥ 18 kcal/kg/day by day 3 has been associated with significantly higher odds of 180-day readmission [35,36,37]. Similar patterns have been observed in oncology populations, where in-hospital dietary intake was not consistently associated with readmission outcomes, likely due to the overriding influence of disease burden and ongoing treatment effects [27]. These findings indicate that nutritional interventions alone—even when individualized and protocol-driven—may be insufficient to reduce readmissions without integration into multidisciplinary transition-of-care strategies and robust post-ICU monitoring. Further studies are warranted to clarify how the timing, intensity, and continuity of nutrition therapy affect post-discharge trajectories, especially in patients with complex or chronic conditions. Future research should explore whether combining structured nutritional strategies with coordinated transitional care planning can help mitigate readmission risks in high-risk populations. Causes of hospital readmission were not adjudicated; therefore, the higher early readmission rate should be interpreted as hypothesis-generating rather than causal.

### 4.4. Strengths and Limitations

This study possesses several notable strengths. First, it leveraged a large, real-world ICU cohort with diverse clinical characteristics, enhancing the generalizability of the findings. Second, the structured protocol implemented in the DGIN group enabled frequent, individualized nutritional assessments by clinical dietitians. Patients in the DGIN group received an initial evaluation within 48 h of ICU admission, followed by two scheduled reassessments during the first week and three additional follow-up evaluations from the second week onward (timing flexible; Figure 2). This represents a pragmatic yet intensive nutritional care model that may be feasible in other high-acuity settings. Third, the study employed multivariable adjustments for key clinical and nutritional confounders—including BMI, APACHE II score, energy and protein intake, albumin, and potassium—thereby improving the robustness of between-group comparisons.

However, several limitations should be acknowledged. The observational design of this study precludes definitive conclusions regarding causality. Because prescribed energy/protein targets were not recorded longitudinally at the patient level, adherence to individualized targets and target-versus-achieved discrepancies could not be evaluated in this retrospective dataset. Body weight during critical illness is an insensitive marker of nutritional status due to fluid shifts and acute catabolism; direct measures of body composition (e.g., bioimpedance, CT-derived muscle area, or ultrasound) and cumulative fluid balance were not available, limiting inference about lean-mass preservation. While multivariable adjustments were applied, residual confounding from unmeasured factors such as ICU discharge policies, comorbidities, functional status, and psychosocial determinants may still have influenced the observed outcomes, particularly readmission rates. Furthermore, defining the “final stable phase” based on the period prior to ICU transfer may introduce variability related to unit-level practices or individual clinician discretion, despite being conducted within a single hospital. The reliance on retrospective nutritional intake documentation may also be subject to reporting bias, although consistency in hospital charting protocols may have mitigated this effect. Finally, although the frequency of dietitian contact was well-defined in the DGIN group, the intensity of standard care in the SC group was less standardized and may have varied depending on staff availability or clinical judgment.

Together, these strengths and limitations provide important context for interpreting the observed associations between structured nutritional intervention and clinical outcomes in critically ill patients. As an observational study, residual and unmeasured confounding cannot be excluded despite matching; therefore, our findings should be interpreted as associations rather than causal effects and warrant prospective confirmation.

## 5. Conclusions

In this single-center retrospective cohort, DGIN was associated with a shorter ICU length of stay compared with SC, without a mortality difference and with higher early hospital readmission (14-/30-day). These findings should be interpreted cautiously and validated prospectively.

## Figures and Tables

**Figure 1 nutrients-17-02995-f001:**
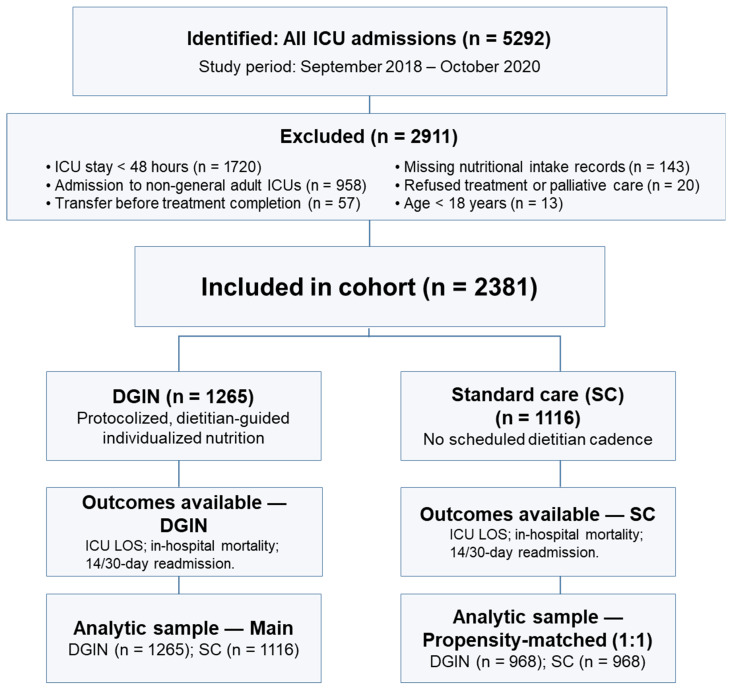
Flowchart of cohort selection and analysis sets. Among 5292 intensive care unit (ICU) admissions between September 2018 and October 2020, 2911 were excluded (for reasons detailed in the flowchart), yielding a cohort of 2381 patients stratified into the dietitian-guided individualized nutrition (DGIN) group (n = 1265) and the standard care (SC) group (n = 1116). A 1:1 propensity score–matched subset was created for sensitivity analysis (n = 968 per group). Abbreviations: DGIN, dietitian-guided individualized nutrition; SC, standard care; ICU, intensive care unit.

**Figure 2 nutrients-17-02995-f002:**
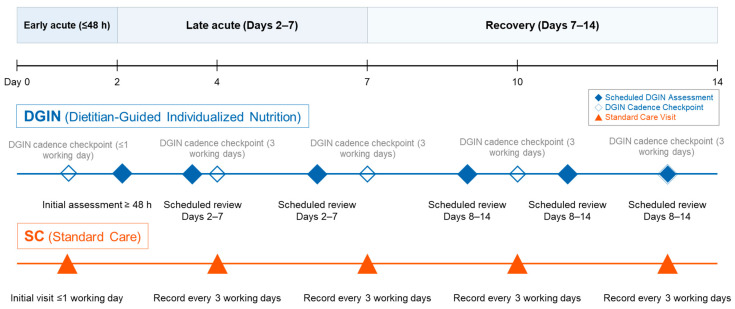
Comparison of the dietitian-guided individualized nutrition (DGIN) protocol and the standard care (SC) cadence for nutritional support. The timeline illustrates two proactive nutritional care models across three clinical phases of the intensive care unit (ICU) stay. The DGIN protocol (top panel) represents an intensified, structured approach. It includes an initial assessment within 48 h of admission and a high-frequency schedule of subsequent reviews during the late acute (days 2–7) and recovery (days 7–14) phases, with built-in cadence checkpoints to ensure a minimum contact frequency. The SC pathway (bottom panel) represents the standard institutional cadence, consisting of an initial visit within one working day followed by routine documentation every three working days. Markers denote scheduled DGIN assessments (circles), DGIN cadence checkpoints (diamonds), and SC visits (triangles).

**Figure 3 nutrients-17-02995-f003:**
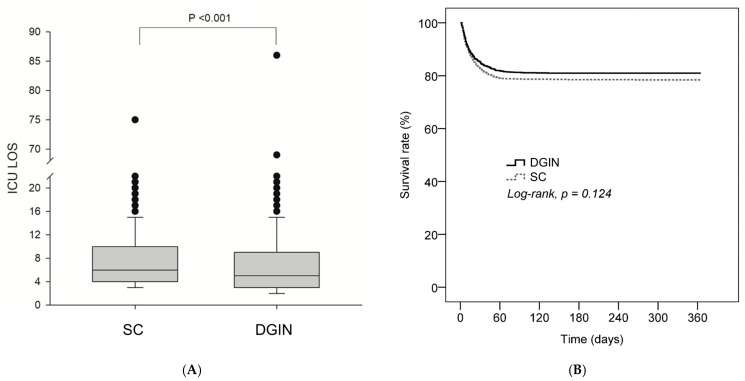
(**A**) Boxplot comparing ICU length of stay (LOS) between the SC and DGIN groups. The DGIN group had a significantly shorter ICU LOS (*p* < 0.001). (**B**) Kaplan–Meier survival curves showing no significant difference in survival probability between the groups (log-rank *p* = 0.124).

**Table 1 nutrients-17-02995-t001:** Baseline characteristics at ICU admission (SC vs. DGIN).

Variable	SC (*n* = 1116)	DGIN (*n* = 1265)	*p* Value
Age, years	66.0 ± 16.4	65.7 ± 16.6	0.629
Sex, Male, *n* (%)	432 (38.7)	488 (38.6)	0.947
Admission type, *n* (%)			0.247
- Medical	678 (60.8)	739 (58.4)	
- Surgical	438 (39.2)	526 (41.6)	
Weight, kg	60.9 ± 14.0	62.5 ± 15.3	0.021
BMI, kg/m^2^	23.5 ± 6.7	24.3 ± 9.1	0.005
APACHE II score	18.3 ± 7.0	17.3 ± 7.3	0.004
Energy at ICU admission, kcal/kg/day	14.7 ± 11.1	13.4 ± 10.5	0.005
Protein at ICU admission, g/kg/day	0.6 ± 0.5	0.5 ± 0.4	<0.001
Laboratory Values			
Albumin, g/dL	3.1 ± 0.6	3.2 ± 0.7	0.001
Prealbumin, mg/dL	13.8 ± 7.2	13.3 ± 7.8	0.557
Hemoglobin, g/dL	10.1 ± 2.1	10.0 ± 2.0	0.620
BUN, mg/dL	36.9 ± 37.7	36.9 ± 36.1	0.749
Creatinine, mg/dL	1.9 ± 2.1	1.9 ± 2.2	0.818
Potassium, mmol/L	3.8 ± 0.8	3.9 ± 0.7	0.012
Calcium, mg/dL	8.2 ± 0.8	8.2 ± 0.8	0.642
Magnesium, mg/dL	2.1 ± 0.4	2.1 ± 0.4	0.558
Phosphorus, mg/dL	4.0 ± 1.9	4.1 ± 2.0	0.380
C-reactive protein, mg/L	6.2 ± 6.9	6.4 ± 7.1	0.781

Values are mean ± SD or *n* (%). *p* values are from the *t* test or Mann–Whitney U test for continuous variables and the χ^2^ test or Fisher’s exact test for categorical variables. Calculations are based on available cases. Abbreviations: ICU, intensive care unit; SC, standard care; DGIN, dietitian-guided individualized nutrition; BMI, body mass index; BUN, blood urea nitrogen.

**Table 2 nutrients-17-02995-t002:** Clinical outcomes—overall and propensity-matched cohorts (SC vs. DGIN).

Outcome	SC	DGIN	Adjusted OR (95% CI) ^†^	*p* Value
**Panel A. Overall cohort**				
In-hospital mortality, *n* (%)	241/1116 (21.6)	241/1265 (19.1)	aOR 0.99 (0.75–1.30)	0.955
14-day readmission, *n* (%)	90/752 (12.0)	97/401 (24.2)	aOR 2.25 (1.52–3.32)	<0.001
30-day readmission, *n* (%)	90/659 (13.7)	97/302 (32.1)	aOR 3.32 (2.23–4.95)	<0.001
ICU length of stay, days (mean ± SD)	8.1 ± 6.7	7.1 ± 7.4	—	<0.001
**Panel B. Propensity-matched cohort (1:1)**				
In-hospital mortality, *n* (%)	198/968 (20.5)	199/968 (20.6)	aOR 1.05 (0.79–1.40)	0.123
14-day readmission, *n* (%)	73/664 (11.0)	77/306 (25.2)	aOR 2.42 (1.61–3.63)	<0.001
30-day readmission, *n* (%)	84/588 (14.3)	70/229 (30.6)	aOR 3.05 (2.01–4.62)	<0.001
ICU length of stay, days (mean ± SD)	8.1 ± 6.8	7.4 ± 7.6	—	<0.001

^†^ Adjusted for BMI, APACHE II score, energy delivery, protein intake, albumin, and potassium. Continuous variables are presented as mean ± SD; *p* values for continuous outcomes were calculated using the independent *t* test or Mann–Whitney U test, as appropriate; categorical variables are shown as n (%) and compared using χ^2^ or Fisher’s exact test. Matched results were derived from propensity-score matching (PSM). “Readmission” refers to hospital readmission (any inpatient unit) among patients discharged alive from the index hospitalization.

## Data Availability

Data supporting the results of this study are available from the corresponding author upon reasonable request. Data are not publicly available due to institutional policies and patient privacy regulations.

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
