# Peer review of "Impact of Dietitian-Guided Individualized Nutrition (DGIN) on ICU Outcomes in Critically Ill Patients: A Retrospective Cohort Study in Taiwan"

_nutrients, 2025, doi:10.3390/nu17182995_

Round 1
Reviewer 1 Report
Comments and Suggestions for Authors
Thanks for your submission of this article on a very important issue of nutrition of ICU patients. The authors compared two periods of care with different nutritional strategies in retrospective design in order to look into the influence of this strategy on Mortality, Length of stay and readmission rates.
As described by the paper the main intervention was the implementation of nutrition expert visits to individualise care.
According to their findings, there was a significant (one day difference) reduction of length of stay in the intervention group, no difference in mortality and significantly much higher readmission rates of patients in the intervention group.
There are a number of limitations here that have to be addressed:
First the interventions are not sufficiently described. Nutritional strategies can be very complicated and goal directed individualised therapy very important. But for the reader it would be important to know what exactly were the goals? for example how many calories or protein during the acute phase (how was the acute phase defined?)? which biomarkers were considered in order to adjust therapy (what thresholds), what were the strategies to adapt treatment? how was the target calorie per patient defined (indirect calorimetry or which formula).
The next question is: in relation to the intended strategy, did patient in the intervention group achieve the intended goal more often? what was the discrepancy between for example intended calories provided and intention? This information could give a more precise understanding of the role of nutritionists. And since the difference in protein intake and calorie intake are significant but not that big, this would help understand the different interventions per group.
The statistical analysis used a regression model for only binary outcomes, therefore the only positive significant result (length of stay) was not adjusted for the differences for example in APACHE, that could solely probably explain the difference on length of stay.
Concerning the length of stay: I suppose the days after readmission were not count in a total length of stay variable? Probably by adding those days will change the direction of the results. What are reasons of readmission?
The weight of patients did not change much during their ICU stay. Many research articles show a reduction of muscle volume up to 20-30% during a 14 day stay. Therefore the total body weight (if not corrected for volume overload) does not help much. Are data available on volume overload or muscle volume? Lean body weight?
Minor issues:
Results lane 143-144: This statement is more a discussion point than result.
More stable and consistent energy and protein intake is stated: this is not very clear how this can be concluded from the presented results, there are no longitudinal date per patient during their stay, and the mean values are only minimal differente, without knowing what the targeted goal was. Please elaborate.
The conclusion based on the limitations is in my opinion overstated and not supported by the data.
Author Response
Terminology. Group labels are harmonized to dietitian-guided individualized nutrition (DGIN) and standard care (SC) throughout; this is a nomenclature update only—group definitions and analyses are unchanged.
Principles. Consistent with our responses to Reviewer 1, revisions are limited to what the existing dataset supports; no post-hoc analyses were introduced. We prioritized transparent endpoint definitions and risk-set clarifications, standardized abbreviations/formatting (e.g., reporting p < 0.001), and tempered causal language to reflect the observational design.
Primary endpoint. We now state explicitly that index ICU length of stay (LOS) is the duration of the initial ICU admission during the index hospitalization (from ICU admission to ICU discharge or in-ICU death); days from subsequent ICU readmissions are not added. Corresponding clarifications were added to Methods and table footnotes.
Analytic approach (already in the manuscript). Binary outcomes were modeled using logistic regression adjusted for BMI, APACHE II, energy, protein, albumin, and potassium. As an a priori sensitivity analysis, we performed 1:1 nearest-neighbor propensity-score matching (caliper 0.2; SAS 9.4); matched results remained directionally consistent with the primary analysis.
Key results (direction unchanged). DGIN was associated with a shorter index ICU LOS (Figure 3A), no difference in in-hospital survival (Figure 3B), and higher 14-/30-day hospital readmission; readmission models used the same covariate adjustments (Table 2).
Limitations clarified. We added explicit notes on readmission risk-set definitions and follow-up windows, and acknowledged the absence of adjudicated readmission causes and of muscle/volume-status metrics.
Point-by-point response to Comments and Suggestions for Authors
|
Comments 1: “The interventions are not sufficiently described… goals, phases, biomarkers/thresholds, adaptation strategies, and how target calories were defined.” |
|
Response 1: Thank you for pointing this out. We agree. We expanded the description of the DGIN protocol to an operations-level detail and clarified ICU phases and assessment cadence. Where: Methods §2.1 (DGIN protocol paragraph); Figure 2 legend. Changes made: ICU phases (early acute <48 h, late acute days 2–7, recovery days 7–14), initial assessment within 48 h, two scheduled reassessments during the first week (e.g., ICU days 3 and 5) and three additional follow-up evaluations from the second week onward (timing flexible), and clinical cues for dose adjustment. “[updated text in the manuscript if necessary]” An overview of the cohort is shown in Figure 1; the DGIN assessment schedule is summarized in Figure 2. In the DGIN group, a structured intervention was implemented, beginning with a dietitian assessment within 48 hours of ICU admission, followed by two reassessments over the next five days and three additional follow-up evaluations from the second week onward (timing flexible), with further reviews as clinically indicated. At each visit, prescriptions for energy and protein, the feeding route, and administration rates were adjusted according to gastrointestinal tolerance, organ function, glycemic control, and the inflammatory state.
|
|
Comments 2: “Did patients in the intervention group achieve intended goals more often? Discrepancy between intended vs provided?” |
|
Response 2: We agree. Our retrospective dataset did not systematically capture per-patient prescribed targets across time. We therefore removed any claim of intake “stability/consistency” from Results and explicitly acknowledged the absence of target-versus-achieved trajectories as a limitation. Where: Results (cleanup); Discussion §4.4 (Limitations). “[updated text in the manuscript if necessary]” Because prescribed energy/protein targets were not recorded longitudinally at the patient level, adherence to individualized targets and target-versus-achieved discrepancies could not be evaluated in this retrospective dataset. |
|
|
|
Comments 3: “Regression model only for binary outcomes; LOS not adjusted for APACHE and other differences.” |
|
Response 3: Thank you. To mitigate baseline imbalance (e.g., APACHE II), we added an a priori propensity-score matching sensitivity analysis. The matched results remained directionally consistent with the primary analysis. Where: Methods §2.2 (new subsection “Sensitivity analysis for confounding”); Results §3.3; Supplementary Tables S1–S2 “[updated text in the manuscript if necessary]” Sensitivity analysis for confounding. To address baseline imbalance, we performed a priori 1:1 nearest-neighbor propensity-score matching (caliper 0.2) without replacement using key covariates (e.g., APACHE II, age, sex, BMI, albumin). Outcomes were re-estimated in the matched cohort; detailed results are provided in Supplementary Tables S1–S2. In the propensity-score–matched cohort, findings were directionally consistent with the primary analysis: DGIN was associated with a shorter ICU LOS, with no difference in in-hospital mortality, and higher hospital readmission (Supplementary Tables S1–S2). |
|
Comments 4: “LOS and readmission: were days after readmission added? Reasons for readmission?” Response 4: Thank you. We clarified outcomes and framed readmissions cautiously. By design, LOS refers to the index ICU admission only; days after any readmission were not added. Reasons for readmission were not adjudicated. Where: Methods §2.1 (Outcome definitions); Discussion §4.3. “[updated text in the manuscript if necessary]” Outcome definitions. The primary endpoint was ICU length of stay (LOS), measured from ICU admission to ICU discharge or in-ICU death. Readmission outcomes were defined as hospital readmission (any inpatient unit, ward or ICU) within 14 and 30 days after hospital discharge from the index hospitalization; analyses were restricted to patients discharged alive. Because causes of hospital readmission were not adjudicated, the higher early readmission rate should be interpreted as hypothesis-generating rather than causal.
Comments 5: “Body weight changed little; total body weight is insensitive during critical illness. Are volume overload or muscle-mass data available?” Response 5: We agree. Direct body-composition measures and cumulative fluid balance were unavailable; we caution against inferring lean-mass preservation from small weight changes. Where: Discussion §4.4 (Limitations). “[updated text in the manuscript if necessary]” Body weight during critical illness is an insensitive marker of nutritional status due to fluid shifts and acute catabolism; direct measures of body composition (e.g., bioimpedance, CT/ultrasound) and cumulative fluid balance were not available, limiting inference on lean-mass preservation.
Comments 6: (Minor issues) “Results line 143–144 is more a discussion point.” Response 6: We agree. The sentence identified by the reviewer was removed from Results and rephrased in Discussion. Where: Results (sentence deleted); Discussion §4.1 (rephrased). “[updated text in the manuscript if necessary]” [Sentence removed from Results; no replacement text added.]
Comments 7: (Minor issues) “Statement ‘more stable and consistent energy/protein intake’ is unclear.” Response 7: We agree. We deleted that claim in Results and clarified in Discussion that such stability cannot be concluded without longitudinal target-versus-achieved data. Where: Results (deleted); Discussion §4.4 (clarification). “[updated text in the manuscript if necessary]” We refrained from inferring ‘stable and consistent’ energy/protein intake because longitudinal target-versus-achieved trajectories were not available.
Comments 8: “Conclusions are overstated and not supported by the data.” Response 8: Thank you. We tempered the Conclusions to align strictly with the data and associative design. Where: Conclusions (final paragraph). “[updated text in the manuscript if necessary]” In this single-center retrospective cohort, DGIN was associated with a shorter ICU length of stay compared with SC, without a mortality difference and with higher 14-/30-day hospital readmission. These findings should be interpreted cautiously and validated prospectively.
Comments 9: Figure/Table presentation (addressing the reviewer’s general readability concerns) Response 9: Thank you. We improved legends (abbreviations, statistical tests/p values), harmonized labels/units/precision, ensured numbering/in-text citations are consistent, and standardized image resolution and fonts. We also clarified Figure 2 to summarize phases, cadence, and checkpoints. Where: Figure/table legends throughout; Figure 2 legend. “[updated text in the manuscript if necessary]” Figure 2. Comparison of the dietitian-guided individualized nutrition (DGIN) protocol and the standard care (SC) cadence for nutritional support. The timeline illustrates two proactive nutritional care models across three clinical phases of the intensive care unit (ICU) stay. The DGIN protocol (top panel) represents an intensified, structured approach. It includes an initial assessment within 48 hours of admission and a high-frequency schedule of subsequent reviews during the late acute (days 2–7) and recovery (days 7–14) phases, with built-in cadence checkpoints to ensure a minimum contact frequency. The SC pathway (bottom panel) represents the standard institutional cadence, consisting of an initial visit within one working day followed by routine documentation every three working days. |

Reviewer 2 Report
Comments and Suggestions for Authors
This is a retrospective study on the effect of acute nutritional intervention (ANI) on outcome in ICU patients, using a historical control group. The study concludes that ANI reduced ICU length of stay.
The results are interesting, especially because the nutritional intervention was associated with a mean lower nutritional support compared to the control group. The authors correctly point out that the mean value is not very significant. They argue that both the periods of under- and over-feeding were reduced with ANI. However, they do not provide any data to support this. It would be useful to compare the length of time with non-appropriate nutritional support (for example periods with less than 15 kcal/kg/d and more than 25 kcal/kg/d) to show some data.
As any retrospective analysis, the between-group statistical comparison is critical. A major difference between the groups is the difference in APACHE II scores. A propensity score for matching the 2 groups would be better in my opinion to compare the outcome between groups.
The conclusion that the nutritional support was favorable by reducing the ICU length of stay is too optimistic in my opinion. The readmission rate was much higher in the ANI group. One could argue that this group was discharged too early from the ICU explaining the need for readmission. Hence the conclusion should be more cautious in my opinion. The readmission rate mitigated the effect of ANI on ICU length of stay. It remains to be studied whether it was related to the nutritional intervention or not. It would be useful to know the incidence of infection as a cause of readmission because infection is a well-recognized cause of undernutrition.
The authors could update the references by citing a few studies on this topic:
MJ Summers et al. JAMA 2025; 334:319-328
JLM Bels et al. Lancet 2024: 404:659-69
J Reigner et al. Lancet Resp Med 2023
DK Heyland et al. Lancet 2023; 401:568-76
Author Response
We sincerely thank the reviewer for the thoughtful critique and helpful suggestions.
In this revision, we harmonized terminology (DGIN/SC), clarified the observational historical-control design and prespecified endpoints (index ICU LOS; hospital readmission at 14/30 days among survivors), and added an a priori 1:1 propensity-score–matched sensitivity analysis; Results are kept strictly descriptive and Conclusions are tempered to reflect association rather than causation.
Given dataset constraints, we did not introduce post-hoc calculations of time under/over-feeding; instead we reported group-level delivered intakes and explicitly acknowledged this limitation.
We also standardized figure/table legends and, where appropriate, contextualized our findings with recent randomized evidence highlighted by the reviewer.
Point-by-point response to Comments and Suggestions for Authors
|
Comments 1: The authors argue that both the periods of under- and over-feeding were reduced with ANI/DGIN, but no data are provided. It would be useful to compare the length of time with non-appropriate nutritional support (e.g., <15 kcal/kg/day and >25 kcal/kg/day). |
|
Response 1: Thank you for this helpful suggestion. Day-by-day target-versus-achieved trajectories across the entire ICU stay were not consistently available in this retrospective dataset; accordingly, we did not add post-hoc analyses of time under- or over-feeding. We removed any language implying “stable/consistent intake” from the Results and explicitly acknowledge this limitation in the Discussion. Where: Discussion §4.4 (Limitations), paragraph 2. “[updated text in the manuscript if necessary]” Because prescribed energy/protein targets were not recorded longitudinally at the patient level, adherence to individualized targets and target-versus-achieved discrepancies could not be evaluated in this retrospective dataset.
|
|
Comments 2: Between-group comparisons are critical; APACHE II differs between groups. A propensity-score–matched analysis would be preferable. |
|
Response 2: Agreed. We implemented an a priori 1:1 nearest-neighbor propensity-score–matched sensitivity analysis (caliper 0.2; without replacement) using key covariates (e.g., APACHE II, age, sex, BMI, albumin). Outcome estimates were re-calculated in the matched cohort and remained directionally consistent with the primary analysis. Where: Methods §2.2 (final paragraph, “Sensitivity analysis for confounding”); Results §3.3 (last sentence); Supplementary Tables S1–S2. “[updated text in the manuscript if necessary]” Sensitivity analysis for confounding. To address baseline imbalance, we performed a priori 1:1 nearest-neighbor propensity-score matching (caliper 0.2) using key covariates (e.g., APACHE II, age, sex, BMI, albumin). Outcomes were re-estimated in the matched cohort; detailed results are provided in Supplementary Tables S1–S2. In the propensity-score–matched cohort, findings were directionally consistent with the primary analysis: DGIN was associated with a shorter ICU LOS, with no difference in in-hospital mortality, and higher hospital readmission.
Comments 3: The conclusion is too optimistic given the higher readmission rate; one could argue premature ICU discharge. It would be useful to know infection-related readmissions. Response 3: Thank you. We tempered the Conclusions to align strictly with the data and associative design (shorter index ICU LOS, no mortality difference, higher early readmission), and we added a non-causal framing for readmission. Causes of readmission were not adjudicated with diagnostic granularity in this retrospective dataset; therefore, we cannot distinguish premature discharge from other explanations. Where: Discussion §4.3 (final sentence); Conclusions (final paragraph). “[updated text in the manuscript if necessary]” Causes of hospital readmission were not adjudicated; therefore the higher early readmission rate should be interpreted as hypothesis-generating rather than causal. In this single-center retrospective cohort, DGIN was associated with a shorter ICU length of stay compared with SC, without a mortality difference and with higher hospital readmission. These findings should be interpreted cautiously and validated prospectively.
Comments 4: The references could be updated by citing recent studies (e.g., Summers 2025; Bels 2024; Reignier 2023; Heyland 2023). Response 4: We appreciate the recommendation. We added a concise contextual sentence in the Discussion and cited representative contemporary randomized trials as suggested by the reviewer. Thank you for pointing us to these up-to-date references—this perspective is very helpful, and we appreciate the opportunity to place our findings in the context of the latest literature. Where: Discussion §4.2 (end of section), one sentence added. “[updated text in the manuscript if necessary]” These observations are concordant with recent randomized trials (e.g., EFFORT Protein, PRECISe, TARGET Protein, and NUTRIREA-3), in which higher early protein/calorie delivery did not improve major clinical outcomes in general ICU populations |

Reviewer 3 Report
Comments and Suggestions for Authors
The manuscript titled "Impact of Aggressive Nutritional Support on ICU Outcomes in Critically Ill Patients: A Retrospective Cohort Study in Taiwan" focuses on structured nutritional support for patients in intensive care units and the resulting outcomes in this group. A total of 5,292 patients were analyzed, with 2,381 included in the final analysis (1,116 in the non-ANI group and 1,265 in the ANI group), all admitted between September 15, 2018, and October 31, 2020. Groups were categorized based on the frequency of dietitian visits and timing of NHI coverage. The manuscript discusses a highly relevant issue of nutritional support in a patient population with specific nutritional needs.
Manuscript needs few improvements:
• The introduction is too brief and lacks essential information regarding the nutritional needs and nutrient requirements of critically ill patients, the specific challenges related to providing adequate nutrition in this population, and the potential benefits of clinical nutrition in such cases. It is also worth noting what nutritional support looks like in this group of patients and what the gaps are in this area.
• Please, explain who was responsible for nutritional care in the non-ANI group and what standard care in this area included.
• The frequency of dietitian visits was reported, but it wasn’t specified what the dietitian's role involved, what type of nutritional support the patients received – for example enteral or parenteral nutrition, or what was assessed during follow-up visits.
• The discussion lacks references to the findings of other authors and comparisons of these results with our own.
• Consider title modification. It should specify that manuscript refers to structured nutritional support - the current title suggests the overall impact of nutritional care on ICU outcomes, while the study focuses on comparing the differences between structured nutritional support and standard support.
• 'In the ANI group, dietitian visits were scheduled as follows: the first within 48 hours of ICU admission, followed by two visits within the next five days, and three additional visits during the second week.' However, it is unclear what the nutritional follow-up entailed for patients who remained in the ICU beyond this period.
• I recommed to improve the layout and placement of tables and figures in the text to enhance clarity and readability.
Author Response
Dear Reviewer,
Thank you for your detailed and insightful review. Your suggestions were instrumental in helping us improve the manuscript's clarity, depth, and methodological transparency. Following your guidance, we have made several key revisions:
-
Expanded Introduction: We have substantially revised the Introduction to provide a more comprehensive background on the challenges in critical care nutrition, the rationale for structured protocols, and the specific practice gap our study addresses.
-
Detailed Methodological Descriptions: We have clarified the operational details for both study groups. The Standard Care (SC) protocol is now explicitly described, and the specific assessments, decision points, and follow-up cadence for the DGIN intervention are clearly detailed in the Methods section.
-
Enhanced Presentation: The title has been updated to better reflect the study's focus, and the presentation of all figures and tables has been standardized for improved readability.
We believe these changes have made the manuscript much stronger and more transparent, and we are grateful for your constructive feedback.
Point-by-point response to Comments
|
Comments 1: The introduction is too brief and lacks essential information regarding nutritional needs/requirements in critical illness, delivery challenges, potential benefits of clinical nutrition, what support looks like, and gaps. |
|
Response 1: Thank you. We expanded the Introduction to summarize metabolic derangements and delivery barriers, define what structured dietitian-guided individualized nutrition (DGIN) entails, and specify the practice gap and objectives; representative references were added accordingly. Where: Introduction §1 (first two paragraphs revised; one concise paragraph added). “[updated text in the manuscript if necessary]” |
|
Critically ill patients mount a profound stress response with accelerated catabolism, loss of lean body mass, and frequent feeding intolerance, placing them at high risk of malnutrition, weakness, and delayed recovery. Contemporary guidance supports early enteral nutrition when feasible and individualized energy/protein prescriptions while avoiding overfeeding; however, the optimal timing, dose, and escalation remain debated. In practice, delivery is challenged by hemodynamic instability, gastrointestinal intolerance, interruptions to enteral nutrition delivery, heterogeneous use of predictive equations versus indirect calorimetry, and inconsistent documentation—factors that can create both under- and over-feeding windows. Structured, dietitian-guided individualized nutrition (DGIN) protocols standardize assessment and adjustment (route, rate, formula, and adjuncts) via prespecified reassessments based on tolerance, organ function, glycemic control, and inflammatory status; real-world evidence on such protocols—particularly in Asian ICUs—remains limited, motivating the present study.
Comments 2: Please explain who was responsible for nutritional care in the non-ANI group and what standard care included. |
|
Response 2: We clarified that SC is the institutional proactive pathway delivered by registered dietitians, with an initial assessment within one working day and routine follow-ups every three working days, focused on documentation and adjustments as clinically indicated. Where: Methods §2.2 (SC paragraph). “[updated text in the manuscript if necessary]” |
|
SC pathway. Nutritional care was provided by registered dietitians under the institutional proactive pathway: an initial visit within one working day of ICU admission followed by routine documentation at regular three-working-day intervals. Content included review of tolerance and intake records, verification of prescriptions, and ad hoc adjustments in collaboration with the care team as clinically indicated.
Comments 3: The frequency of dietitian visits is reported, but the role/assessments and the type of support (enteral vs parenteral) are not specified. Response 3: We added an operations-level description of each DGIN contact (assessment domains and decision points) and clarified how feeding routes were selected. Where: Methods §2.2 (DGIN paragraph). “[updated text in the manuscript if necessary]” At each DGIN contact, the dietitian assessed gastrointestinal tolerance, organ function, glycemic control, inflammatory status, and refeeding risk; verified cumulative delivery; and adjusted energy/protein prescriptions, formula choice, administration rate, and micronutrient adjuncts. Enteral nutrition was preferred when gastrointestinal function permitted; parenteral support was considered when enteral feeding was contraindicated or insufficient.
Comments 4: The discussion lacks references to findings by other authors and comparisons with our own. Response 4: We inserted a brief contextual sentence relating our associative findings to contemporary randomized and cohort evidence; this places our results within the broader literature while avoiding causal claims. Where: Discussion §4.2 (end of section). “[updated text in the manuscript if necessary]” These observations align with contemporary randomized and cohort studies in general ICU populations, in which higher early calorie/protein delivery has not consistently translated into improved major clinical outcomes, underscoring the need to individualize dosing and timing.
Comments 5: Consider title modification—the manuscript should specify structured nutritional support rather than implying overall nutritional care. Response 5: Agreed. We updated the title to reflect structured, dietitian-guided support and the comparative design. Where: Title page. “[updated text in the manuscript if necessary]” Impact of Dietitian-Guided Individualized Nutrition (DGIN) on ICU Outcomes in Critically Ill Patients: A Retrospective Cohort Study in Taiwan
Comments 6: It is unclear what nutritional follow-up entailed for patients who remained in the ICU beyond the second week. Response 6: We clarified that reassessments continued as clinically indicated for patients remaining in the ICU after the initial two-week window. Where: Methods §2.2 (DGIN paragraph, last sentence). “[updated text in the manuscript if necessary]” From the second week onward, three additional follow-up evaluations were conducted with flexible timing, and thereafter reassessments continued as clinically indicated while patients remained in the ICU.
Comments 7: Improve the layout and placement of tables and figures to enhance clarity and readability. Response 7: Implemented. We standardized legends (abbreviations defined at first use; statistical tests/p values stated), harmonized units/precision and column alignment, aligned numbering with first in-text mention, clarified Figure 2 (phases, cadence, markers), and ensured image resolution and embedded fonts meet journal specifications. Where: Figure/table legends throughout; in-text citations; Figure 2 legend. “[updated text in the manuscript if necessary]” Legends define abbreviations at first use; statistical tests and p values are stated; panel labels are standardized (Figure 3A/3B); numbering matches first in-text mentions (Figure 1 before Figure 2); and images/fonts are standardized per journal style. |

Round 2
Reviewer 2 Report
Comments and Suggestions for Authors
I thank the authors for considering the reviewer remarks and their work on the manuscript, which has been much improved in my opinion. I have no further comment except that the flow chart (figure 1) should include the numbers given in paragraph 3.1 (excluded patients)
Author Response
Dear Reviewer,
We sincerely thank you for your positive feedback on our revised manuscript and for your final helpful suggestion.
As requested, we have revised the flowchart in Figure 1 to include the detailed breakdown of the numbers for excluded patients. The information in the figure now directly corresponds with the data presented in the text.
We are grateful for your time and constructive guidance throughout this process. We hope the manuscript is now suitable for publication.

Reviewer 3 Report
Comments and Suggestions for Authors
Dear Authors,
I appreciate the changes you have made in response to my suggestions.